# Combinations of Cannabinoids with Silver Salts or Silver Nanoparticles for Synergistic Antibiotic Effects against Methicillin-Resistant *Staphylococcus aureus*

**DOI:** 10.3390/antibiotics13060473

**Published:** 2024-05-22

**Authors:** John Jackson, Ali Shademani, Manisha Dosanjh, Claudia Dietrich, Mark Pryjma, Dana M. Lambert, Charles J. Thompson

**Affiliations:** 1Faculty of Pharmaceutical Sciences, University of British Columbia, 2405 Wesbrook Mall, Vancouver, BC V6T 1Z3, Canada; 2Department of Biomedical Engineering, University of British Columbia, 2222 Health Sciences Mall, Vancouver, BC V6T 1Z3, Canada; 3Department of Microbiology and Immunology, University of British Columbia, Vancouver, BC V6T 1Z3, Canadacthompso@mail.ubc.ca (C.J.T.); 4Andira Pharmaceuticals Inc., 1600-925 W Georgia Street, Vancouver, BC V6C 3L2, Canada

**Keywords:** cannabinoids, silver, synergy, MRSA

## Abstract

Silver has been shown to improve the antibiotic effects of other drugs against both Gram- positive and -negative bacteria. In this study, we investigated the antibiotic potential of cannabidiol (CBD), cannabichromene (CBC) and cannabigerol (CBG) and their acidic counterparts (CBDA, CBCA, CBGA) against Gram-positive bacteria and further explored the additive or synergistic effects of silver nitrate or silver nanoparticles using 96-well plate growth assays and viability (CFUs- colony-forming units). All six cannabinoids had strong antibiotic effects against MRSA with minimal inhibitory concentrations (MICs) of 2 mg/L for CBG, CBD and CBCA; 4 mg/L for CBGA; and 8 mg/L for CBC and CBDA. Using 96-well checkerboard assays, CBC, CBG and CBGA showed full or partial synergy with silver nitrate; CBC, CBDA and CBGA were fully synergistic with silver nanoparticles against MRSA. Using CFU assays, combinations of CBC, CBGA and CBG with either silver nitrate or silver nanoparticles, all at half or quarter MICs, demonstrated strong, time-dependent inhibition of bacterial growth (silver nitrate) and bactericidal effects (silver nanoparticles). These data will lead to further investigation into possible biomedical applications of specific cannabinoids in combination with silver salts or nanoparticles against drug-resistant Gram-positive bacteria.

## 1. Introduction

As natural products, *Cannabis sativa* extracts have a long history of medicinal use against numerous diseases despite little scientific information about individual cannabinoid activities or possible mechanisms of action. With the downregulation of marijuana from an illegal narcotic to a legal or decriminalized material in many countries, the study of the pharmacology of cannabinoids is growing rapidly.

More than one hundred different cannabinoids have been identified from *C. sativa*. The well-known and most studied phytocannabinoids, cannabidiol (CBD) and delta 9- tetrahydrocannabinol (THC), are known to interact with CB1 and CB2 receptors, which make up the well-established endocannabinoid system in humans. In this system, CB1 is located mainly in the CNS and CB2 is located mainly in the immune system [1,2]. The strong binding of THC to CB1 produces its psychotropic effects. CBD weakly associates with both CB1 and CB2 and is thought to counterbalance some of the psychotropic effects of THC via this more indirect interaction with CB1 [1,2]. Other phytocannabinoids present in *C. sativa* include cannabichromene (CBC) and cannabigerol (CBG), with their relative concentrations varying significantly among chemovars [3]. All four of these cannabinoids (THC, CBD, CBG and CBC) are products of their respective acid forms (THCA, CBDA, CBGA and CBCA) [3]. Both CBG [3] and CBC [4] are reported to bind to only CB2 (not CB1) but there is some controversy around whether these agents have any downstream effect on these receptors.

Much of the research on cannabinoids in the 1950s and 1960s used alcoholic extracts from plants like hemp and identified some antibiotic activity of these cannabinoid extracts, particularly CBD, CBC and CBG, against Gram-positive bacteria [5,6]. More recent studies found that CBD in hemp oil extracts has no Gram-negative antibiotic activity but good activity against three Gram-positive species (*Enterococcus* spp., *Staphylococcus* spp. and *Listeria* spp.) at concentrations in the low µg/mL range [7]. Similarly, CBC, CBD and CBG all inhibited the growth of dental plaque bacteria; further, cannabis seed extract gave very strong inhibition of *Staphylococcus aureus* growth and minor inhibition of representative Gram-negative bacteria (*E. coli* and *Pseudomonas aeruginosa*) [8]. Recently, Kosgadage et al. [9] reported that CBD gave small increases in the bactericidal activity of several established antibiotics against Gram-negative bacteria, suggesting that cannabinoids may have a supportive role in infection management.

Generally, it is now well accepted that cannabinoids alone do not have significant antibacterial effects against Gram-negative bacteria; however, they are active against some drug-resistant strains of Gram-positive bacteria. Chakraborty et al. [10] showed that ethanol extracts of *C. sativa* had strong activity against methicillin-resistant *Staphylococcus aureus* (MRSA) which could be amplified by the addition of other natural product extracts like Thuya by unknown mechanisms. In a detailed review, Appendino et al. [11] identified CBD, CBC, CBG and THC as effective antibiotics against MRSA. Similarly, Farha et al. [12] highlighted CBG as a strong antibiotic against MRSA biofilms and suggested that this cannabinoid targets the cytoplasmic membrane of the bacteria. The study also reported that CBG could be effective against Gram-negative bacteria if the outer membrane was permeabilized, suggesting this cannabinoid might target the inner membrane.

These data on the antibiotic effects of cannabinoids identified three particular agents (CBD, CBG and CBC) as having good activity against Gram-positive bacteria and being potential anti-infective agents against MRSA and MRSA biofilms. Interestingly, the acidic forms of CBD and CBG have been reported to have similar effects but at approximately 50% lower activities, whereas the acidic form of CBC (CBCA) is reported to have a faster and more potent effect than vancomycin against MRSA [13].

Unlike cannabinoids, silver has an ancient history of use as an antibiotic against many types of bacteria and infections. With the discovery of modern antibiotics like penicillin in 1942, the use of silver fell out of favor. However, currently, with the rise of drug resistance against nearly all modern antibiotic compounds, silver is attracting renewed attention, largely because the silver agents may inhibit multiple targets so that resistance is less likely than for single target-specific drugs [14,15,16]. Silver is particularly attractive as an antibiotic due to its low toxicity to human cells and its antibacterial effect against both Gram-positive and -negative bacteria. Silver nitrate has a long history of use in the prevention and treatment of wound infection and is effective against *Staphylococcus aureus*, *E. coli* and *Pseudomonas aeruginosa* [15]. Silver binds to proteins, induces bacterial membrane damage and binds to both DNA and RNA to inhibit replication [15]. The use of silver sulfadiazine topicals and silver-impregnated wound dressings like Acticoat^TM^, (Smith and Nephew, Mississauga, ON, Canada) which contains silver nanoparticles, and Aquacel^TM^ (ionic silver) (Convatec, Paddington, London, UK), which forms a gel on wound contact, is now well established in developed countries [17,18]. Silver nitrate has largely given way to the use of silver nanoparticles, which stay intact and bind to bacterial membranes as well as many sites within the cell [15,16]. Of particular interest, silver (as silver nitrate) enhances the antibiotic action of ampicillin, gentamicin and ofloxacin against Gram-negative bacteria like *E. coli* and also sensitizes such cells to Gram-positive-specific antibiotics like vancomycin [14]. Similarly, in checkerboard studies, silver nanoparticles were shown to improve the antibiotic effect of ampicillin, chloramphenicol and kanamycin against Gram-positive (*Enterococcus* spp., *Staphylococcus* spp. and *Streptococcus* spp.) as well as against Gram-negative (*E. coli* and *P. aeruginosa*) bacteria [19]. Clearly, there is great interest in combining the known activities of silver nitrate or silver nanoparticles with existing antibiotics to produce additive or synergistic antibiotic effects. Such research may allow for the restoration of bacterial sensitivity to previously effective antibiotic agents to which there is resistance. Since previous reports point to CBD, CBG and CBC as having antibiotic activity against Gram-positive bacteria, the objective of this study was to investigate the synergistic or additive action of these cannabinoids with silver nitrate or silver nanoparticles against MRSA.

## 2. Results

### 2.1. Cannabinoid–Silver Checkerboard Assays

All six cannabinoids (CBD, CBC and CBG and their acidic counterparts (CBDA, CBCA, and CBGA) had antibacterial activity against MRSA with MICs below 10 mg/L as shown in Figure 1a,b and Table 1. CBD, CBG and CBCA had the strongest inhibitory activity, all with MICs of 2 mg/L. In combination with silver nitrate over a range of concentrations, 96-well plate checkerboard assays showed full synergy (FICI < 0.5) for CBC and partial synergy (FICI < 0.75) for CBG and CBGA (Table 1). In combination with silver nanoparticles, CBC, CBDA and CBGA each showed synergistic antibiotic effects, with FICIs of 0.14, 0.25 and 0.375, respectively (Table 1).

### 2.2. Cannabinoid–Silver Synergy Studies Using Colony-Forming Unit Counts

**CBC.** The MIC of silver nitrate with MRSA was 16 mg/L so it was used at sub MIC concentrations of 8, 5 and 1 mg/L in these synergy studies. Using both half and quarter MICs of CBC (4 and 2 mg/L, respectively) in combinations with silver nitrate at 8 and 5 mg/L allowed for rapid bactericidal activity against MRSA (i.e., >99.999% reduction of a 1 × 10^6^ CFU inoculum after 2 h), which persisted for the 24 h duration of the experiment (Figure 2c), whereas 8 mg/L silver nitrate alone produced bacteriostatic growth inhibition over 6 h, after which CFU increased to match that of the no-treatment group at 24 h. For silver nanoparticles, similar effects were observed. There was no antibacterial effect of silver nanoparticles alone at 1, 2.5 or 5 mg/L; however, rapid, bactericidal effects (>99.999% CFU reduction) were observed for 24 h with 5 mg/L silver nanoparticles in combination with CBC at half the MIC (Figure 2b). Rapid bactericidal activity also occurred with 1 mg/L silver nanoparticles in combination with CBC at half the MIC; however, the effect was less pronounced (99.9% CFU reduction at 24 h). Similar but less pronounced effects were observed using CBC at one-quarter of the MIC. There was no significant difference in effects between 10 nm and 20 nm nanoparticles (Figure 2a,b).

**CBGA**. When bacteria were incubated with CBGA at 2 mg/L (half the MIC) together with silver nitrate at 1 mg/L, there was a clear increase in the antibiotic effect (Figure 3c). There was no antibiotic effect observed for silver nanoparticles (either 10 nm or 20 nm) alone up to 5 mg/L, but a strong antibiotic effect for all nanoparticle concentrations used in combination with CBGA at either 2 mg/L or 1 mg/L (Figure 3a,b). There was no difference in effect between 10 nm and 20 nm nanoparticles.

**CBG.** There was an improved antibiotic effect of CBG at 1 mg/L (half MIC) in the presence of silver nitrate at 1 mg/L (Figure 4c). The combination of silver nanoparticles (20 nm) at 5, 1 or 0.25 mg/L with CBG at 1 mg/L resulted in a 100% antibiotic effect. The addition of 5 mg/L silver nanoparticles also resulted in an increased antibiotic effect (Figure 4b) for CBG at 0.5 mg/L (quarter MIC). Similar but less pronounced results were observed for 10 nm silver nanoparticles. The combination of silver nanoparticles at 5 and 1 mg/L with CBG at 1 mg/L resulted in a full antibiotic effect (Figure 4a), but for CBG at one-quarter of the MIC, only the addition of silver nanoparticles at 5 mg/L allowed for an improved antibiotic effect and that only transiently (2–6 h).

**CBD.** There was a small inhibition of bacterial growth observed using CBD at 1 mg/L (half MIC) with silver nanoparticles at 5 mg/L or silver nitrate at 1 mg/L (Figure 5a,b).

**CBCA.** There was full inhibition of bacterial growth in the presence of CBCA at 1 mg/L with silver nanoparticles at 5 mg/L (Figure 6a), but partial inhibition of bacterial growth using 1 mg/L CBCA (half MIC) with silver nitrate at 1 mg/L (Figure 6b).

**CBDA.** There was little inhibition of bacterial growth with CBDA at 8 or 4 mg/L as seen in Figure 7. The mild inhibition of growth observed with higher concentrations of AgNP or silver nitrate was not enhanced in the presence of CBDA. 

In all these CFU studies, there was little difference between effects observed with 10 nm or 20 nm. It should be noted that the results section for CFU counts largely refers to the 24 h time point in each study. However, a more dramatic antibiotic effect of cannabinoids in combination is often observed at time points up to 6 h in Figure 2, Figure 3, Figure 4, Figure 5 and Figure 6.

### 2.3. PVA Film Studies

Over the six-hour period of incubation, control bacteria grew to well over 10^8^ CFU from the starting concentration of 5 × 10^5^, as seen in Figure 8 and Figure 9. CFU data are reported as the mean ± the st deviation on *n* = 4 samples. The inclusion of silver nanoparticles at wt% concentrations (to PVA) between 0.003 and 0.025% (i.e., 1.5–12.5 µg) inhibited bacterial growth so that CFU values dropped to approximately 10 < 7 CFU (Figure 8 and Figure 9).

The inclusion of CBG at up to approximately 0.025% (12.5 µg) concentrations had minor inhibitory effects on MRSA CFU values (Figure 8). However, the combined inclusion of silver nanoparticles (range 0.012 to 0.025%, i.e., 6–12 µg) with CBG (range 0.006 to 0.012%, i.e., 3–6 µg) resulted in almost full inhibition of bacterial growth for the following combinations (%silver nanoparticles + %CBG): 0.012 + 0.012, 0.025 + 0.006 and 0.025 + 0.012 (Figure 8).

The inclusion of CBC in the 0.003 to 0.012% concentration range (% of PVA, i.e., 1.5 µg–6 µg CBC) partially inhibited bacterial growth from CFU values between 10 < 8 and 10 < 9 down to values close to 10 < 7 (Figure 9). However, the combined inclusion of silver nanoparticles (0.006%, i.e., 3 µg) with CBC (range 0.006 to 0.012%, i.e., 3–6 µg) in the PVA films at all combined concentrations allowed for full inhibition of bacterial growth with values often well under the starting 5 × 10^5^ CFU (Figure 9).

## 3. Materials and Methods

### 3.1. Materials

Lysogeny broth (LB) was obtained from Gibco (Gibco, Thermo Fisher, Nepean, ON, Canada). Silver nitrate (purity greater than 99%) and all cannabinoids (pure reference standards at 1 mg/mL in methanol, purity greater than 99%) were obtained from Sigma-Aldrich (Oakville, ON, Canada). Silver nanoparticles at 20 µg/mL (10 nm or 20 nm) were obtained from Sigma-Aldrich and those at 1 mg/mL (10 nm) were obtained from Nanocomposix (San Diego, CA, USA). All plasticware was obtained from Corning Life Sciences (Union City, CA, USA). All silver nanoparticles were stabilized in sodium citrate buffer and all sizes are means ± 2 nm.

Poly (vinyl alcohol) (selvol 540: 88% hydrolyzed with a molecular weight of 150,000 and selvol 125: 99% hydrolyzed with a molecular weight of 125,000 was purchased from Sekisui Specialty Company (Dallas, TX, USA).

### 3.2. Bacterial Strains and Growth

MRSA, strain USA300, was cultured in LB and inoculated at 37 °C.

### 3.3. 96-Well Plate Checkerboard Assays

Cannabinoids or silver (expressed as the concentration of silver, not the salt) were serially diluted 2-fold across the 96-well plate (Costar, Corning) followed by the addition of 100 µL of bacterial culture with an OD_600_ of 0.0025. Cannabinoid concentrations ranged from 16 mg/L to 0.125 mg/L, silver nitrate ranged from 32 mg/L to 0.31 mg/L and silver nanoparticles ranged from 10 mg/L to 0.01 mg/L. In some 96-well plates, silver nanospheres (Nanocomposix) were used at much higher concentrations of 40 or 80 mg/L with dilutions down from there. Subsequently, plates were wrapped with aluminum foil and incubated for 24 h. The turbidity in each well was then analyzed using a Varioskan microplate reader (Thermo Fisher, Nepean, ON, Canada).

### 3.4. FICI Computation

The Fractional Inhibitory Concentration Index (FICI) was used to explore how the antibacterial effect of one drug is enhanced by the addition of a second drug (in combination) by creating a matrix of combined drug concentration. FICI was calculated in a checkerboard assay based on the turbidity of the wells. The FIC of each agent was determined as the ratio of the minimal inhibitory concentration (MIC) of one agent in the presence of the other agent to the MIC of that agent alone. The FICI was consequently computed as the sum of each agent’s FIC. Note: FIC indices (FICI) were interpreted as follows: ≤0.5, synergy; <0.5–≤0.75, partial synergy; 0.75–≤1.0, additive effect; >1.0–≤4.0, indifference; and >4.0, antagonism as similarly described by others [20,21].

### 3.5. Colony-Forming Unit, Kill-Curve Test

Once the antibacterial profiles of combinations of drugs were determined at one time point (24 h) using FICI, colony-forming unit assays were used to study the time course of the combined antibacterial effects of two agents in more detail by measuring the numbers of viable bacterial cells rather than the turbidity of the cultures in the 96-well plates. For each test tube, 1 mL of culture at OD_600_ of 0.005 was added to 1 mL of LB medium containing antibiotics to reach the target sub-MICs of each compound. Samples of 100 µL were extracted from each tube at the determined time points, followed by 10-fold serial dilutions. Then, 10 µL of each dilution was added onto the LB agar plates that were subsequently incubated for 24 h. Eventually, colonies were counted and the results presented in log CFU/mL. All experiments were repeated on three separate occasions and points are shown as the mean ± the standard deviation (st dev). Note: The errors are very small and often hidden under the marker symbols.

### 3.6. PVA Film Manufacture (Solvent-Cast PVA) and Bacteria Incubation

PVA was prepared as concentrated solutions (10% *w*/*w*) by adding PVA to a known volume of rapidly stirred water at 85–90 °C and then stirring and heating for one hour. The clear solution was cooled to room temperature. Solutions of both PVA forms (99% and 88%) were diluted down to 5% *w*/*w* and mixed together at the ratio of 55:45 (88% hydrolyzed to 99% hydrolyzed). Finally, appropriate amounts of cannabinoids or silver sulfate were added to the blended PVA solutions to give various concentrations of cannabinoids or silver nanoparticles as a percent (*w*/*w*) of total PVA. Then, 50 mg films were cast in 20 mL flat-bottomed glass bottles. The solutions in the glass jars were left in a dark 37 °C oven for 48 h in order for water to evaporate.

For bacterial assays, 1 mL of culture containing 5 × 10^5^ CFU growing in LB medium was added to the PVA film, which was enough to fully swell the film with only a tiny excess of liquid (<25 µL) on top. After 6 h of incubation at 37 °C, the bacteria were extracted by the addition of 2 mL of LB medium and 30 s vortexing to break up the hydrogel film. The number of bacteria remaining was assessed using the colony-forming kill test method as described above.

## 4. Discussion

The antibacterial effects of hemp and cannabis plant extracts have been known for many centuries, but it was not until the 1950s that the cannabinoids CBC, CBG and CBD were clearly identified as powerful antibiotics against Gram-positive bacteria. However, due to the worldwide restrictions on cannabis research in laboratories, the science of these agents is still poorly understood, but the exciting news from more recent studies is that these agents are effective against drug-resistant bacteria at low concentrations with MICs below 10 µg/mL [12]. Two very recent excellent reviews on the antibiotic activities of various cannabinoids against Gram-positive bacteria are Mahmud et al. [22] and Schofs et al. [23].

The growing threat of antibiotic resistance and the lack of pharmaceutical research leading to the discovery of new antibiotics has led to the search for additive or synergistic effects by using combinations of existing antibiotics. In particular, silver has been shown to be effective against drug-resistant bacteria and to improve the antibiotic activity of numerous drugs against many types of bacteria [14,19].

Unfortunately, neither silver nor cannabinoids are approved for permanent internal use, but both agents are suitable for non-permanent device coatings. Silver is already used as an antibiotic on catheter coatings (Bardex^R^) (Bard, New Providence, NJ, USA) and in numerous wound dressings where it is often used in the nanoparticle form (e.g., Acticoat^TM^), so these types of devices would be well served by the inclusion of a second antibiotic agent that augments the action of the first.

In this study, we explored the antibiotic activity of CBC, CBG and CBD (along with their acidic forms) against MRSA bacteria. In agreement with other researchers [5,6,7,8,10,11,12], all six cannabinoids had good activity against MRSA with MICs below 10 µg/mL (Figure 1a,b). The order of antibacterial strength and the MICs (Table 1) were consistent with those described by others [12]. Silver was also included in these studies and in the ionic form (silver salt) had reasonable activity against MRSA (MIC approx. 16 µg/mL) and very low activity in the nanoparticulate form (MIC approx. 80 µg/mL).

The use of 96-well plate checkerboard studies is well recognized as a simple method to identify possible synergy combinations. This allows for the determination of FICI scores which, in very broad terms, indicate synergy if the value is less than 0.5. However, many researchers further define FICI scores with synergy at >0.5, partial synergy at >0.75 and additive at >1 [21,24]. In these studies, CBC, CBG and CBGA displayed full synergy or partial synergy with silver nitrate, and CBC, CBDA and CBGA showed full synergy with silver nanoparticles (Table 1). This is the first report to our knowledge of any antibiotic synergy between cannabinoids and silver. Other researchers have reported that CBD increases the activity of some antibiotics against Gram-negative [9] bacteria or used *Cannabis sativa* extracts in combination with other plant extracts against MRSA [10]. Wassmann et al. [25] showed synergy between CBD and bacitracin against MRSA USA 300, although the MIC of this agent alone was very high (64 µg/mL) and bacitracin is the only FDA-approved for use in poultry or topically in humans.

To further explore the improved antibiotic effects of combinations of cannabinoids with silver, time-dependent colony-forming unit assays were performed for all agents. For CBC, a greatly improved antibacterial effect was observed using CBC with silver nitrate at both half or quarter MICs (Figure 2c), with similar results observed using silver nanoparticles at 5 µg/mL or lower (Figure 2a,b). Similarly, silver nitrate and nanoparticles enhanced the antibacterial effects of both CBG and CBGA, although to a lesser degree than seen for CBC (Figure 3 and Figure 4). Improvements in the antibacterial activity of both CBD and CBCA were also observed for both forms of silver (Figure 5 and Figure 6). Smekalova et al. reported slightly lower MICs for smaller 8 nm silver nanoparticles as compared to 25 nm in a range of Gram-positive and Gram-negative bacteria, but no consistent difference in FICI synergy values when used with penicillin, gentamicin or colistin [26]. Similarly, in these CFU studies, 10 nm and 20 nm silver nanoparticles were used, but no difference in improved antibiotic effect with the addition of cannabinoids was observed between the two.

Silver salts or nanoparticles are well known to have additive or synergistic effects with established antibiotics against both Gram-positive and -negative bacteria [14,19,27]. However, the mechanism of action of silver as an antibiotic alone appears to be multifactorial, so it is unknown how silver enhances the action of other drugs. Briefly, silver ions bind to DNA and RNA to inhibit replication, bind to many proteins and induce bacterial membrane damage, whereas silver nanoparticles bind to bacterial membranes, disrupting permeability and protein function, and they can also enter bacteria to disrupt respiratory chain processes [14,15,16].

Despite its history of use in modern infection control products such as antimicrobial wound care and medical device coatings, Cochrane systematic reviews report no clear evidence that silver products effectively prevent or treat infection, clinically underscoring the need for innovation and improved antimicrobial technologies.

Based on previous works that point to the cannabinoid antibacterial mechanism involving disruption of the inner bacterial membrane [12], the mechanism of synergistic anti-MRSA effects observed between silver and cannabinoids in the present study is likely to result from dual-targeted disruption of bacterial membrane integrity, supported by the rapid onset of bactericidal activity observed.

Rapid bactericidal activity with limited or no bacterial re-growth diminishes the possibility of bacterial resistance emerging; therefore, the enhanced bactericidal effects of cannabinoids in combination with silver may help limit the emergence of resistance. Furthermore, Farha et al. [12] have reported no detectable spontaneous resistance to CBG and no change in the MIC of CBG against MRSA upon 15-day serial passages and sequential subculture experiments, pointing to a very low propensity for the emergence of bacterial resistance.

Polyvinyl alcohol is a biocompatible polymer used in wound dressings, where it forms a comfortable rinsible hydrogel that may be loaded with antibiotics like silver [28]. We have recently shown that PVA films manufactured using a 60:40 ratio of 88%:99% hydrolyzation polymers degrade slowly over a week [28]. In this study, PVA (55:45) films were loaded with extremely low concentrations of either CBG, CBG or silver nanoparticles and incubated with MRSA in broth, enough to allow for complete hydrogel swelling, which might occur in a wound setting with exudate. The bacteria were allowed to grow within the film, as might happen in a hydrogel dressing above a wound, and the effect of the inclusion of very low concentrations of CBG, CBC or silver nanoparticles on bacterial growth was studied at six hours. Because the swelling allowed full hydration of the PVA film, as might happen on a wound, we did not perform in vitro drug release studies using a large excess of aqueous media, as this scenario would never happen on a wound. The 55:45 ratio dressing was chosen as this degrades slightly faster than the 60:40 ratio dressing and may be easily sloughed off a wound site with a saline rinse and then replaced to absorb new exudate. Whilst the inclusion of silver nanoparticles or CBC or CBG alone produced a partial inhibition of bacterial growth, the combination of either cannabinoid with silver nanoparticles at the same concentrations as alone produced much larger inhibitory effects. It should be noted that in a 50 mg film, the inclusion of cannabinoid or silver nanoparticles at a concentration of 0.012% equates to a total load of 6 micrograms of agent in the film or a little over 1 microgram per square centimeter of film. Obviously, these amounts of agents are very low but support the synergistic effect of the combined use of silver nanoparticles with cannabinoids. For clinical application, the concentrations of any of these agents could be safely increased, thereby reducing cost and any safety concerns.

In conclusion, these studies confirm the antibiotic activity of CBG, CBC and CBD and the acidic forms of these agents against drug-resistant Gram-positive bacteria. The addition of silver, either as salts or nanoparticles, to select cannabinoids allows for much improved and, in some cases, synergistic antibiotic activity. Collectively, these findings strongly support further investigation of cannabinoid-enhanced silver preparations to assess their antimicrobial spectrum of activity and potential application in wound dressings or catheter coatings for an extended and more powerful antibiotic profile.

## Figures and Tables

**Figure 1 antibiotics-13-00473-f001:**
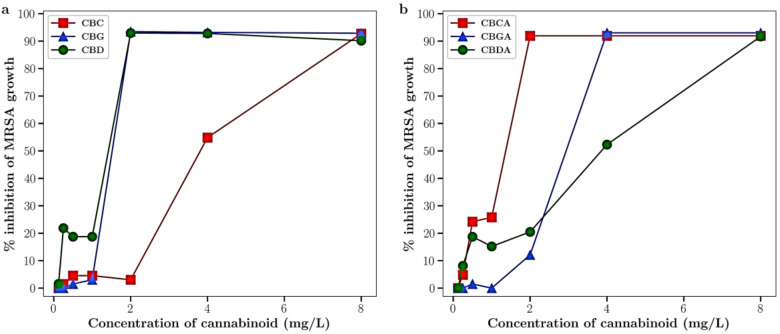
Effect of cannabinoids on the growth of MRSA. (**a**) CBC, CBG and CBD; (**b**) CBCA, CBGA and CBD. *n* = 3 points show mean ± st dev (very small and often hidden under the marker).

**Figure 2 antibiotics-13-00473-f002:**
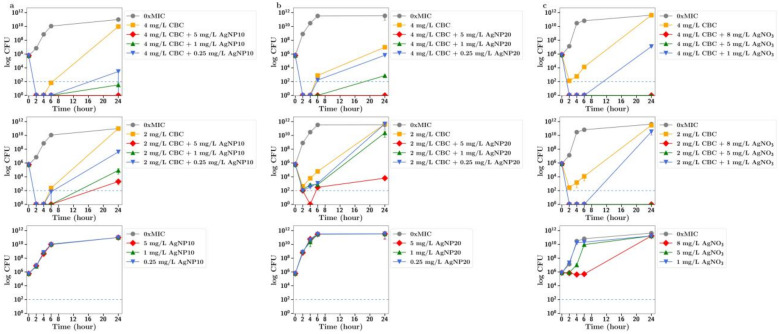
Effect of CBC at half and quarter MICs (4 mg/L, 2 mg/L) on the growth of MRSA in the presence of (**a**) silver nanoparticles (10 nm, AgNP10) at 0.25, 1 or 5 mg/L; (**b**) silver nanoparticles (20 nm, AgNP20) at 0.25, 1 or 5 mg/L; or (**c**) silver nitrate at 1, 5 and 8 mg/L. 0xMIC means Control, no drug.

**Figure 3 antibiotics-13-00473-f003:**
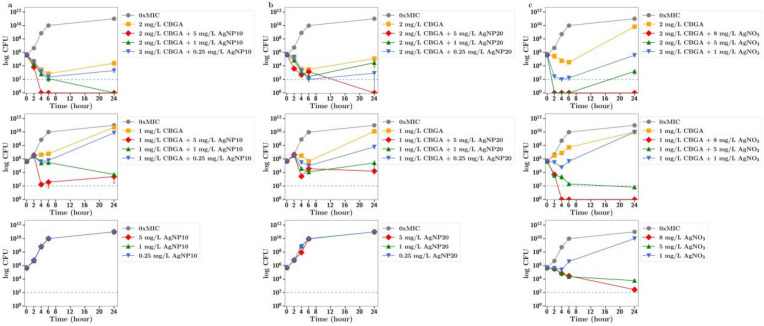
Effect of CBGA at half and quarter MICs (1 mg/L, 2 mg/L)) on the growth of MRSA in the presence of (**a**) silver nanoparticles (AgNP10) at 0.25, 1 or 5 mg/L; (**b**) silver nanoparticles (20 nm, AgNP20) at 0.25, 1 or 5 mg/L; or (**c**) silver nitrate at 1, 5 and 8 mg/L. 0xMIC means Control, no drug.

**Figure 4 antibiotics-13-00473-f004:**
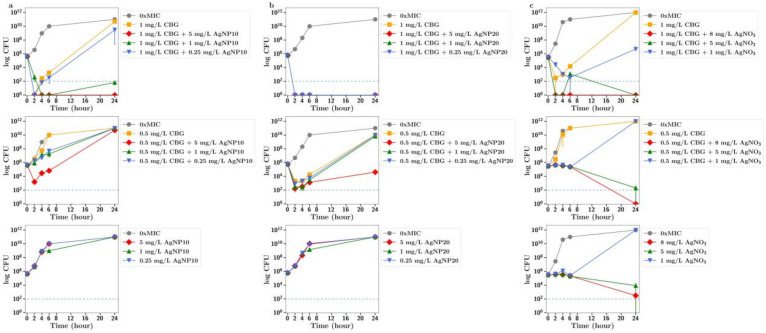
Effect of CBG at half and quarter MICs (1 mg/L or 0.5 mg/L) on the growth of MRSA in the presence of (**a**) silver nanoparticles (AgNP10) at 0.25, 1 or 5 mg/L; (**b**) silver nanoparticles (20 nm AgNP20) at 0.25, 1 or 5 mg/L; or (**c**) silver nitrate at 1, 5 and 8 mg/L. 0xMIC means Control, no drug.

**Figure 5 antibiotics-13-00473-f005:**
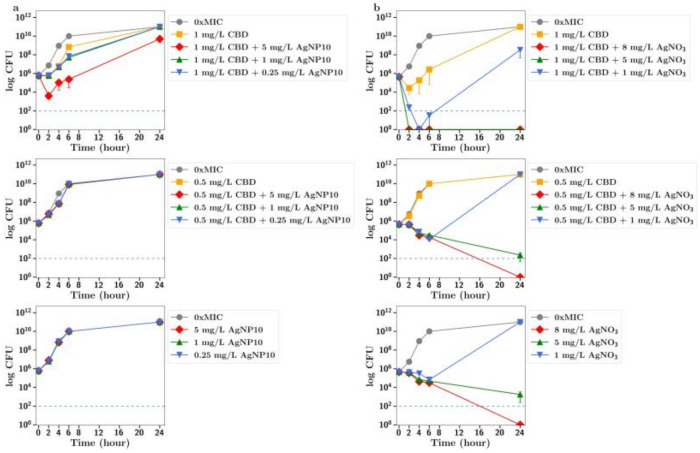
Effect of CBD at half and quarter MICs (1 mg/L or 0.5 mg/L) on the growth of MRSA in the presence of (**a**) silver nanoparticles (10 nm, AgNP10) at 0.25, 1 or 5 mg/L or (**b**) silver nitrate at 1, 5 or 8 mg/L. 0xMIC means Control, no drug.

**Figure 6 antibiotics-13-00473-f006:**
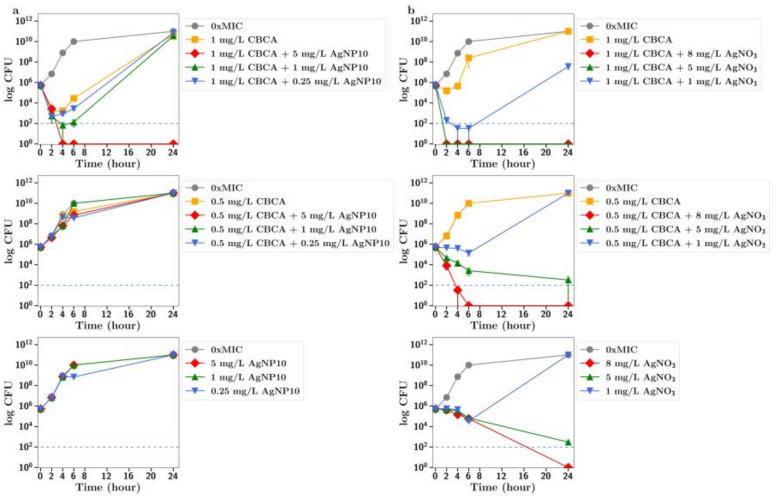
Effect of CBCA at half and quarter MICs (1 mg/L or 0.5 mg/L) on the growth of MRSA in the presence of (**a**) silver nanoparticles (10 nm, AgNP10) at 0.25, 1 or 5 mg/L or (**b**) silver nitrate at 1, 5 and 8 mg/L. 0xMIC means Control, no drug.

**Figure 7 antibiotics-13-00473-f007:**
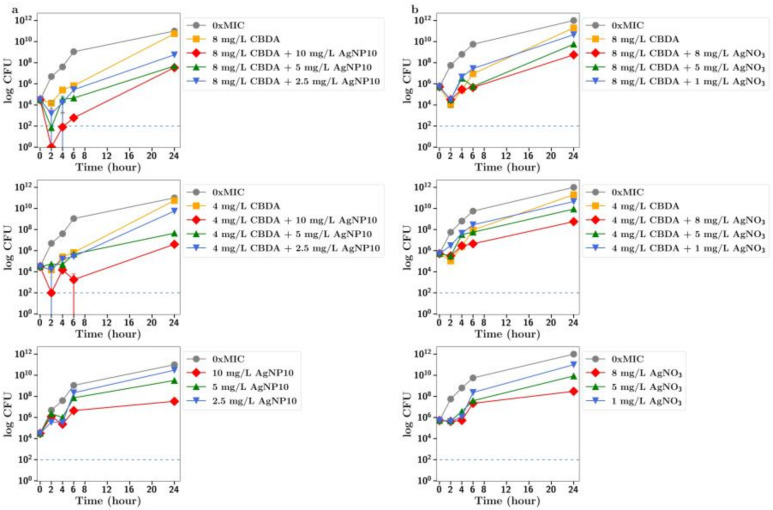
Effect of CBDA at MIC and half the MIC on the growth of MRSA in the presence of (**a**) silver nanoparticles (10 nm, AgNP10) at 0.25, 1 or 5 mg/L or (**b**) silver nitrate at 1, 5 and 8 mg/L. 0xMIC means Control, no drug.

**Figure 8 antibiotics-13-00473-f008:**
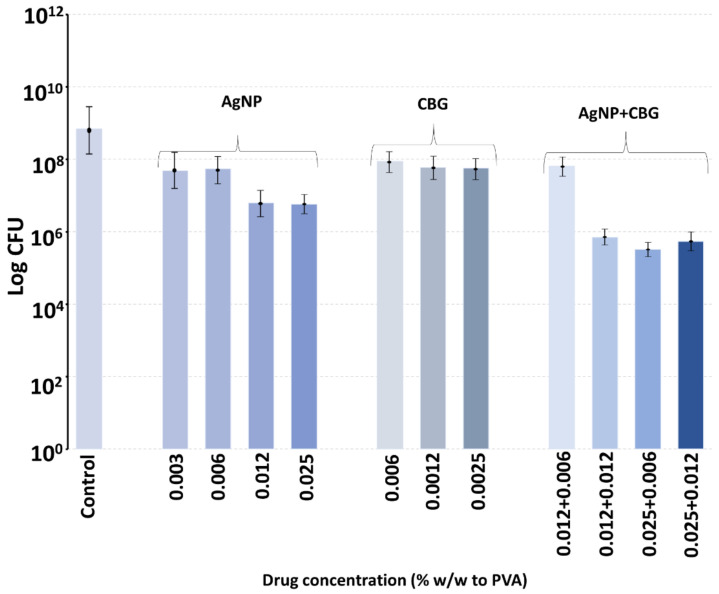
Inhibition of MRSA bacterial growth at 6 h in PVA hydrogel films containing various combinations of AgNP and CBG concentrations following inoculation with 5 × 10^5^ bacteria.

**Figure 9 antibiotics-13-00473-f009:**
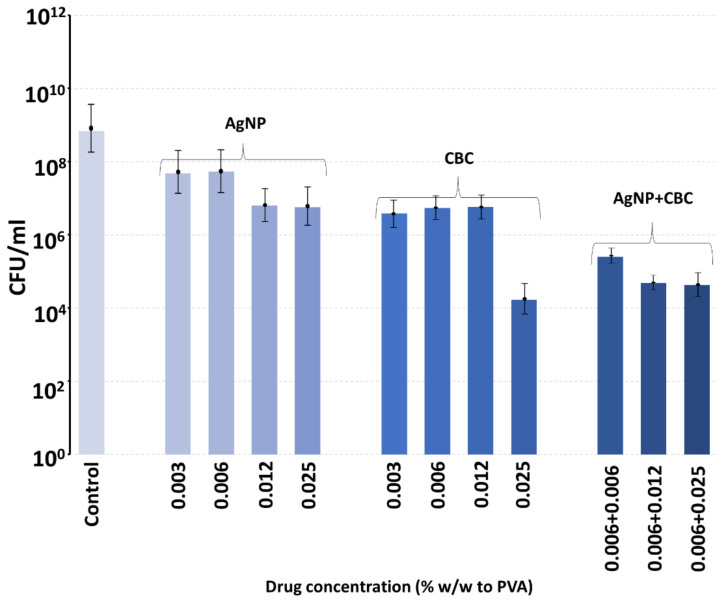
Inhibition of MRSA bacterial growth at 6 h in PVA hydrogel films containing various combinations of AgNP and CBC following inoculation with 5 × 10^5^ bacteria.

**Table 1 antibiotics-13-00473-t001:** Minimal inhibitory concentrations (MICs) of cannabinoids and Fractional Inhibitory Concentration Indices (FICIs) of silver nanoparticles (10 nm; AgNP) or nitrate (AgNO_3_) in combination with cannabinoids in MRSA (USA300).

Cannabinoid	MIC	FICI	
		AgNP	AgNO_3_
CBC	8	0.14	0.375
CBG	2	0.625	0.625
CBD	2	2	2
CBCA	2	2	2
CBGA	4	0.375	0.53
CBDA	8	0.25	2

## Data Availability

The raw data supporting the conclusions of this article will be made available by the authors upon request.

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
