# Peer review of "Combinations of Cannabinoids with Silver Salts or Silver Nanoparticles for Synergistic Antibiotic Effects against Methicillin-Resistant Staphylococcus aureus"

_antibiotics, 2024, doi:10.3390/antibiotics13060473_

Round 1

Reviewer 1 Report

Comments and Suggestions for Authors

The paper discusses the antibacterial properties of cannabinoids (such as THC, CBD, CBC, and CBG) from Cannabis sativa, particularly against gram-positive bacteria like MRSA. It highlights the growing interest in cannabinoids due to changes in legal status in many regions. The study specifically explores the enhanced antibacterial effects when cannabinoids are combined with silver salts or nanoparticles, demonstrating synergistic effects that could lead to new antibacterial treatments, particularly for antibiotic-resistant bacteria. The research suggests potential applications in medical devices like wound dressings and catheter coatings.

Here are some question the author should address:

1,Is there any evidence or speculation within the study about the potential for bacteria to develop resistance to these cannabinoid-silver combinations? How might this compare to resistance development for traditional antibiotics?

2.What differences, if any, were observed between the use of silver salts and silver nanoparticles in terms of their synergistic effects with cannabinoids? Which form of silver showed the most promise in enhancing the antibacterial activity of cannabinoids?

Comments on the Quality of English Language

The English language quality of your manuscript is generally good, with clear and well-structured sentences that convey complex scientific information effectively. However, there are a few areas where improvements could enhance readability and precision:

  1. Consistency in Terminology: It is advisable to maintain consistent terminology throughout the document when referring to cannabinoids and their properties. This includes consistent use of either the chemical names or abbreviations but not interchangeably without clear definitions.

  2. Grammar and Syntax: Minor grammatical errors and awkward phrasings occasionally disrupt the flow. These could be streamlined with the help of a thorough proofread focusing on verb tense consistency and subject-verb agreement.

  3. Technical Jargon and Clarity: While the technical language is appropriate for the subject matter, consider adding brief explanations or definitions for highly specialized terms. This adjustment would make the paper more accessible to readers who are not specialists in this particular area of pharmacology or microbiology.

  4. Punctuation and Sentence Structure: Attention to punctuation, especially the use of commas and semicolons, can improve the clarity of complex sentences. Breaking down overly long sentences into simpler constructions could also aid understanding without sacrificing detail.

  5. Use of Passive and Active Voice: There's a mix of passive and active voices throughout the text. While passive voice is common in scientific writing, using active voice can make statements more direct and dynamic, particularly in discussing the research findings and future steps.

  6. References and Citations: Ensure that all references are cited consistently within the text and that the citation style complies with the journal’s requirements. Inconsistencies in reference formatting can distract from the content and give a less professional appearance to the manuscript.

Reviewer 2 Report

Comments and Suggestions for Authors

The manuscript is entitledCombinations of cannabinoids with silver salts or silver nanoparticles for synergistic antibiotic effects against methicillin resistant Staphylococcus aureus”. This study examined the antibiotic properties of cannabidiol (CBD), cannabichromene (CBC), and cannabigerol (CBG), as well as their acidic forms (A), against Gram-positive bacteria. Additionally, the study investigated whether the combination of silver nitrate or silver nanoparticles had an additive or synergistic effect. The researchers used 96-well plate assays and CFU counting methods to analyze the results. All six cannabinoids exhibited potent antibacterial activity against MRSA. Through the utilization of 96-well plate assays, it was shown that CBC, CBG, and CBGA exhibited complete or partial synergy when combined with silver nitrate. Additionally, CBC, CBDA, and CBGA demonstrated full synergy when combined with silver nanoparticles against MRSA. The CFU tests showed that the combinations of CBC, CBGA, and CBG with both silver nitrate and silver nanoparticles effectively inhibited bacterial growth (silver nitrate) and killed bacteria (silver nanoparticles) in a time-dependent manner. The work is well-written and well-organized. However, there are a few problems that need to be addressed.

The method should specify the percent purity of all cannabinoids and silver nitrate.

Please explain the reason why MRSA strain USA300 was selected for the study.

The standard antibiotic should be used as a positive control.

The manuscript reported only the concentration of the PVA film manufactured. The amount of each cannabinoid in the film should be reported in mg or µg units as well.

The reference for the synthesis of hydrogel films based on PVA should be provided.

The statistical analysis methods, such as one-way ANOVA, mean, and standard deviation, should be reported.

The resolution quality of figures 2 to 7 should be improved.

There are numerous errors in typing such as lines 51, 54, 62, 111, 136, 156, 158, 242, 283.

Comments on the Quality of English Language

There are numerous errors in typing.

Author Response

Reviewer 2.

The manuscript is entitled “Combinations of cannabinoids with silver salts or silver nanoparticles for synergistic antibiotic effects against methicillin resistant Staphylococcus aureus”. This study examined the antibiotic properties of cannabidiol (CBD), cannabichromene (CBC), and cannabigerol (CBG), as well as their acidic forms (A), against Gram-positive bacteria. Additionally, the study investigated whether the combination of silver nitrate or silver nanoparticles had an additive or synergistic effect. The researchers used 96-well plate assays and CFU counting methods to analyze the results. All six cannabinoids exhibited potent antibacterial activity against MRSA. Through the utilization of 96-well plate assays, it was shown that CBC, CBG, and CBGA exhibited complete or partial synergy when combined with silver nitrate. Additionally, CBC, CBDA, and CBGA demonstrated full synergy when combined with silver nanoparticles against MRSA. The CFU tests showed that the combinations of CBC, CBGA, and CBG with both silver nitrate and silver nanoparticles effectively inhibited bacterial growth (silver nitrate) and killed bacteria (silver nanoparticles) in a time-dependent manner. The work is well-written and well-organized. However, there are a few problems that need to be addressed.

The method should specify the percent purity of all cannabinoids and silver nitrate.

Response:  We have now included the purities in the Methods section.

Please explain the reason why MRSA strain USA300 was selected for the study.

Response:  The MRSA strain USA300 is now accepted as the standard from used by laboratories around the world for MRSA studies.

The standard antibiotic should be used as a positive control

Response:  Because the effects of numerous antibiotics against MRSA USA300 are so well characterized by others we felt if unnecessary to include a positive control drug .  The objective of these studies was not to compare cannabinoid/silver antibiotic potential against established drugs but to investigate silver-cannabinoid combination effects relative to each compound alone.  The low mg/L concentrations of each drug alone match values obtained by other workers against MRSA.     .

The manuscript reported only the concentration of the PVA film manufactured. The amount of each cannabinoid in the film should be reported in mg or µg units as well.

Response:  We have now included this information in the Results section.. 

The reference for the synthesis of hydrogel films based on PVA should be provided.

Response: We have included that reference.

The statistical analysis methods, such as one-way ANOVA, mean, and standard deviation, should be reported.

Response: All figures now include error bars (mean +/- st dev from n=3 experiments or n=4 for Figures 8 and 9).  We have now included this information in the text.  Often the error bar is so small it is lost under the point marker.  We have corrected the text to clarify this.  The CFU studies are log scale on the y axis with huge differences in values between control and drug inhibited values.  At 24 hours, most control CFU counts at 1011 whereas inhibited values are usually below 106.   With these huge differences and tiny error bars, we felt the graphs adequately demonstrated the inhibition of growth without overloading the reader with unnecessary Anova data.

The resolution quality of figures 2 to 7 should be improved.

Response:  We have greatly improved the quality of the figures. 

There are numerous errors in typing such as lines 51, 54, 62, 111, 136, 156, 158, 242, 283.

Response:  Many thanks for spotting the errors.  We have corrected all.

Reviewer 3 Report

Comments and Suggestions for Authors

The aim of this study is to investigate synergistic or additive  action  of CBD,  CBG  and  CBC  with silver nitrate or silver nanoparticles in prepared drug loaded PVA polymeric films. However, there are some major and minor issues which must be explained clearly.

-            There are many citation errors in the text (missing citation (line 62), citations are not in correct format (line 54, 265…) etc.)

-            In was mentioned in the study that PVA films were prepared but there is no information about this in title.

-            There are typos in the text (line 100, 134 etc.)  

-            It was mentioned only in results section that 10 nm and 20 nm silver nanoparticles were used but there is no information about these nanoparticles. The DLS results and electron microscopy images of nanoparticles should be added and detailed information can be added to material section.

-            The information about PVA films were missing. The control tests on PVA films must be conducted (TEM images, drug loading and drug release studies etc.)

-            Figure 8 and 9 captions must be changed (PVA films or PVA hydrogels?)

-            There is no statistics in the study. No results were given as result ± SD however in results and discussions sections the authors explained their results as “there is no difference…, similar results were obtained… , showed and improved effect… etc. The authors should add statistics and explained which test did they used in additional statistic section.

Comments on the Quality of English Language

            There are typos in the text (line 100, 134 etc.)  

Round 2

Reviewer 2 Report

Comments and Suggestions for Authors

none

Reviewer 3 Report

Comments and Suggestions for Authors

The manuscript can be accepted in its current form.